# Effectiveness of Two Different Fluoride-Based Agents in the Treatment of Dentin Hypersensitivity: A Prospective Clinical Trial

**DOI:** 10.3390/ma15031266

**Published:** 2022-02-08

**Authors:** Etleva Qeli, Çeljana Toti, Alessandra Odorici, Elisabetta Blasi, Emiljano Tragaj, Michele Tepedino, Francesco Masedu, Gerta Kaçani, Dorjan Hysi, Agron Meto, Luca Fiorillo, Aida Meto

**Affiliations:** 1Department of Conservative, Faculty of Dental Medicine, University of Medicine, 1005 Tirana, Albania; dorjan.hysi@umed.edu.al; 2Department of Prosthetic, Faculty of Dental Medicine, University of Medicine, 1005 Tirana, Albania; celjana.toti@umed.edu.al (Ç.T.); gerta.kacani@umed.edu.al (G.K.); 3Laboratory of Microbiology and Virology, School of Doctorate in Clinical and Experimental Medicine, University of Modena and Reggio Emilia, Via G. Campi, 287, 41125 Modena, Italy; odorici.alessandra@gmail.com; 4Department of Surgical, Medical, Dental and Morphological Sciences with Interest in Transplant, Oncological and Regenerative Medicine, University of Modena and Reggio Emilia, 41125 Modena, Italy; elisabetta.blasi@unimore.it; 5Department of Dentistry, University of Aldent, 1000 Tirana, Albania; emiljanotragaj@yahoo.com (E.T.); agronmeto@yahoo.com (A.M.); 6Department of Biotechnological and Applied Clinical Sciences, University of L’Aquila, 67100 L’Aquila, Italy; michele.tepedino@univaq.it (M.T.); francesco.masedu@univaq.it (F.M.); 7Academy of Sciences of Abruzzo Region, 67100 L’Aquila, Italy; 8Department of Biomedical and Dental Sciences, Morphological and Functional Images, University of Messina, 98100 Messina, Italy; lfiorillo@unime.it; 9Multidisciplinary Department of Medical-Surgical and Odontostomatological Specialties, University of Campania “Luigi Vanvitelli”, 80121 Naples, Italy; 10Endodontic Clinical Section, School of Dentistry, Department of Biomedical and Neuromotor Sciences, University of Bologna, 40125 Bologna, Italy

**Keywords:** dentin hypersensitivity, pulp tester, Tiefenfluorid^®^, Enamelast^TM^, fluoride varnish

## Abstract

Hyperesthesia is related to increased sensitivity of dental tissues to mechanical, chemical and thermal stimuli. The aim of this prospective clinical trial was to compare the effectiveness of a calcium-fluoride-forming agent (Tiefenfluorid^®^, Humanchemie GmbH, Alfeld, Germany) with that of a fluoride varnish (Enamelast^TM^, Ultradent Inc., Cologne, Germany) in the treatment of dental hyperesthesia in adult patients. In total, 176 individuals (106 females and 70 males, aged 18–59 years old) diagnosed with dental hyperesthesia (DH) were enrolled. The main clinical symptoms were hyperesthesia from coldness and sweetness during chewing; the types of clinical lesions were also determined and recorded. The patients were selected randomly and divided into two groups: (i) the first group of 96 patients was treated with Tiefenfluorid^®^ applied in three appointments at 7-day intervals; (ii) the second group of 80 patients was treated with Enamelast^TM^, applied seven times at 7-day intervals. All the patients were recalled 7 days, 14 days, 1 month, 3 months, and 6 months from the last application. At the baseline and during every follow-up visit, the DH was measured with a pulp tester. A random intercept/random slope model was used to evaluate the effect of the treatment, at various times with respect to the initial diagnosis. Within the limits of the present study, Tiefenfluorid^®^ was more effective than Enamelast^TM^ against DH in that it provided long-lasting results, with a significant improvement still detected at the latest 6-month follow-up.

## 1. Introduction

Dentine hypersensitivity (DH) is a clinical condition that significantly interferes with patients’ quality of life (particularly when talking, drinking, eating, and brushing teeth) and may provide unpleasant sensations [1]. Indeed, DH is characterized by short, sharp pain arising from exposed dentinal tubules in response to stimuli, typically thermal, that cannot be ascribed to any other form of dental defect or disease [2,3]. This process occurs when the dentin is exposed in the oral cavity and the dentinal tubules are open. Dentin exposure in the oral cavity occurs due to loss of enamel on the tooth crown and gum recession (loss of cementum after root exposure in oral cavity) [4,5].

Enamel can be lost as a result of friction (aggressive or incorrect brushing technique), biocorrosion (exogenous and endogenous acids), and tooth grinding caused by parafunctional behaviors and stress. Loss of enamel is present in non-carious lesions due to abfraction, erosion, abrasion, and attrition. Pain from coldness, sourness, sweetness, hotness, and airflow are the chief complaints of patients diagnosed with non-carious lesions [5,6,7]. Gum recession may occur as a result of periodontal diseases, prominent roots, habits that lacerate the gingiva, inappropriate use of dental floss, use of toothpicks, inadequate preparation for crowns, or aggressive and incorrect tooth brushing. Operationally, erosion is considered as the progressive loss of hard dental tissues, due to chemical processes that do not involve bacterial infiltration [8]. As the lesion progresses, the tooth walls form a “V” shape, known as a cuneiform defect [9]. The pathological abrasion, which causes excessive loss of the hard dental substance, can lead to the total removal of the anatomical crown [9]. Lastly, gingival recession is the apical migration of the gingival margin to the cemento-enamel junction; it rarely causes tooth loss, but it may be related to thermal and tactile sensitivity, aesthetic complaints, and a tendency toward root caries [10].

Other causes of hypersensitivity include dental treatments such as cavity preparation and tooth bleaching [11]. The prevalence of DH is about 40% in the adult population in many countries [12,13,14]. Furthermore, more than 70% of patients with periodontal disease have experienced DH [14,15]. According to various studies [16,17,18,19,20,21], DH usually occurs in patients within the age range of 30 to 40 and is more prevalent in females than males. Furthermore, DH occurs more often in the canines and premolars than other teeth [17,19,20,21]. The buccal surfaces are mostly affected. According to the widely accepted hydrodynamic theory proposed by Brannström, M., et al. [22], DH occurs as a result of the dentinal fluid movement inside the dentinal tubules, due to thermal, chemical, or osmotic stimuli reaching the exposed dentin [22]. In turn, this movement stimulates baroreceptors. On these bases and in accordance with Drisko, C.H. [23], a list of preventive measures is recommended by dentists for DH patients, such as dietary changes (decreasing the intake of acid-containing foods), nonaggressive and correct tooth brushing techniques, no excessive flossing, no use of toothpicks, no medium- or hard-bristle toothbrushes, no brushing teeth immediately after intaking acidic foods, and use of desensitizing toothpaste. Drisko, C.H. [23] also recommended dental professionals to avoid over-instrumenting the root surfaces during scaling and root planning, to avoid over-polishing the exposed dentin during stain removal, to ensure correct biological width during restoration placement, and to isolate the gingiva during in-office bleaching to avoid burning.

Many professional products (oxalates, resin-based materials, sodium fluoride varnish, silver fluoride solution, glutaraldehyde, arginine-CaCO_3_, and fluoride varnish) have been used to occlude and seal the exposed dentinal tubules [24,25,26,27]. In addition to these, bioactive glasses, such as sodium, calcium phosphosilicate, or calcium phosphate, are used to favor the formation of apatite hydroxycarbonate on the dentin surface, thus occluding the dentinal tubules [28]. In particular, years ago, a crystallized bioactive glass ceramic, named Biosilicate^®^, was released and proposed as an alternative treatment for DH [29,30]. In contact with the dentinal tubules, its particles allow dentinal occlusion by hydroxyapatite, thus providing a strong bond [29,30]. Even fluoride solutions may be relevant against DH, both when used at low concentration in toothpastes and mouthwashes for daily care and when applied at high concentrations by professional personnel, such as fluoride gels and/or fluoride varnishes [31]. Moreover, fluoride varnishes have the advantage of requiring simple application and having prolonged contact with the demineralized surface of the enamel [32]. The main objective of topical fluoride agents (solutions, gels, varnishes) is the deposition of a physical barrier on the surface of the dentin, as calcium fluoride precipitates on the enamel or inside the dentin tubules [33,34,35]. Eventually, such a precipitate may become the source of fluoride that in turn will allow the formation of fluorapatite [33]. On these bases, a calcium-fluoride-forming agent (Tiefenfluorid^®^, Humanchemie GmbH, Alfeld, Germany) has been proposed for the treatment of DH [36]. Initially developed by Prof. Adolph Knappwost, Universities of Hamburg and Tübingen, Germany, Tiefenfluorid^®^ has been used also for caries prophylaxis or remineralization of fissures [36]. To date, few studies are available on its clinical efficacy.

The aim of this prospective clinical trial was to evaluate the effectiveness of the calcium-fluoride-forming agent (Tiefenfluorid^®^, Humanchemie GmbH, Alfeld, Germany) with respect to a fluoride varnish (Enamelast^TM^, Ultradent Inc., Cologne, Germany) in reducing the DH in adult patients with non-carious lesions. The null hypothesis was that there would be no effect of treatment, timing, and diagnosis of the defects on DH measurements.

## 2. Materials and Methods

### 2.1. Study Design

The protocol of the present study was approved by the Local Ethics Committee (Protocol No. 480/2020). Sample size estimation (G*Power version 3.1.9.2, Franz Faul, Universität Kiel, Kiel, Germany) for a random intercept model with three predictors, considering a medium effect size (0.15 according to Cohen), revealed that at least 120 subjects would be needed to achieve 95% power and 5% type I error.

According to the study design, we compared two groups of patients treated with the two fluoride-based agents, Tiefenfluorid^®^ and Enamelast^TM^, for 6 months. The latter was considered the control group, as we considered it unethical to leave DH patients without treatment for such a long time.

### 2.2. Patient Selection

A total of 176 patients, 106 females (F) and 70 males (M) (aged 18–59 years old), diagnosed with DH, were treated. The patients were selected in chronological order and evaluated on the basis of their complaints, such as sensitivity to coldness, to foods and drinks with acidic pH, to sweets, and during teeth brushing. They gave their written consent after being informed at the beginning of the procedures.

### 2.3. Criteria of Inclusion and Exclusion

No age restriction criteria were set in the enrollment. The inclusion criteria were presence of hard tissue erosion, cuneiform defect, pathological abrasion, or gingival recession. The exclusion criteria considered were dental caries, pulp and periodontal pathologies, post-restorative sensitivity, defective fillings, neuropathies, fractures and cracks of the teeth, and patients with a pacemaker.

### 2.4. Diagnosis of DH

The diagnosis of DH was established taking into consideration the subjective examination (time, duration, intensity, and frequency), the clinical examination (tactile and periodontal control), and the radiological examination (pulp and periodontal diseases). Then, the measurement of sensitivity was recorded using a pulp tester and expressed as μA (NSKR HPS, Fencia, Saint-Denis, Réunion, France), in line with other studies [37]. Such an electrodiagnostic reading provides information on the condition, integrity, and functionality of the entire neurovascular bundle, as well as on the sensitivity of the neural apparatus of the pulp. Using this approach, the electro-sensitivity of the pulp can be measured.

### 2.5. Group Division and Fluoride-Based Agent Application

The patients were randomly allocated into the two groups, using a random number sequence. Each patient contributed two teeth randomly selected among those affected by DH to the study. The type of hard tissue erosion, cuneiform defect, pathological abrasion, or gingival recession, associated with DH, was determined, recorded, and stored for every included tooth.

The selected teeth of each patient were treated with either Tiefenfluorid^®^ solution or Enamelast^TM^ varnish (Ultradent Inc., Cologne, Germany), depending on their allocation to the treatment or control group, respectively, following the manufacturer’s instructions.

The treatment group, which included 96 patients, was treated in three sessions with Tiefenfluorid^®^, at time intervals of 7 days. The dental surfaces of this group were first cleaned with pumice and then isolated with a rubber dam. Tiefenfluorid^®^, which consisted of two distinct solutions, was applied as follows: the first solution was applied with a cotton pellet and left in situ for 60 s. Then, the second solution was applied (no rinsing in-between) with another cotton pellet and left in situ for 5 min (time necessary for its evaporation). The surface was rinsed with water only at the end of the procedure. At this point, the patient could eat immediately after the session, since there was no hardening time to respect. The mechanism of Tiefenfluorid^®^ activity on the surface of loose enamel is presented in Figure 1.

The control group, which included 80 patients, was treated over seven sessions with a fluoride varnish, named Enamelast^TM^ (Ultradent Inc., Cologne, Germany). In this group, the dental surfaces were also first cleaned with pumice and then isolated with a rubber dam. Subsequently, the fluoride varnish was applied using a bristle brush; the treatment was repeated every 7 days. Table 1 provides details on the composition of the two fluoride-based agents used in the present study.

### 2.6. Measurement of the DH Level

After the treatments, all patients were observed at the following intervals: 7 days, 14 days, 1 month, 3 months, and 6 months. During each follow-up visit, the DH was measured and recorded using a pulp tester (NSKR HPS, Fencia, Saint-Denis, Réunion, France). The reliability of the electro-sensitivity measurements was evaluated by performing two repeated measurements at a 2-week interval in 59 patients; the random error was calculated using Dahlberg’s formula.

### 2.7. Statistical Analysis

The operator that performed the statistical analysis was blinded with regard to the treatment modalities. Descriptive statistics were computed for all the variables. A Pearson χ^2^ test was used to compare the gender distribution and the defect diagnosis between the two groups. A random intercept/random slope model was used to evaluate the effect of treatment, time, and defect diagnosis on the measurements of DH. Type I error was set as *p* = 0.05 for all the tests. STATA software (version 14, StataCorp, College Station, TX, United States) was used to perform the analysis. 

## 3. Results

Gender distribution was similar between the two groups (Pearson χ^2^ = 0.697, *p* = 0.404). Similarly, the distribution of the different tissue lesions underlying the hypersensitivity (hard tissue erosion, cuneiform defect, pathological abrasion, or gingival recession) was the same across the two groups (Pearson χ^2^ = 1.043, *p* = 0.594). Table 2 shows the pulp tester measurements in terms of mean and standard deviation (SD) obtained in the treatment and control groups. Unlike the control group, the treatment group showed a relevant and time-related increase, with a ~4-fold augmentation at the 6-month follow-up. The objective measurement of DH with a pulp tester showed good reliability, since the Dahlberg formula revealed a random error of 0.05 μA.

The regression model was highly significant (Wald χ^2^ = 57,878.45, *p* < 0.001), revealing that no effect of defect diagnosis was present, while there was a significant effect of the treatment in both groups, particularly evident over time (Table 3).

In general, the treatment group showed a gradual and time-related improvement in the curve over time, compared to the control group, which showed a lower curve that tended to decrease at the 6-month follow-up (Figure 2, Table 4).

## 4. Discussion

The problem of DH has been a subject of discussion over the years. It occupies an important place in dental diseases, with its continuously increasing incidence [38,39,40]. It has been established that the dentinal fluid can move inside the dentin tubules toward the inside of the pulp or the outside of dentin. In particular, the dentinal fluid flows away from the dentin pulp upon exposure to hypertonic chemical stimuli, evaporation, drying, and cooling [41]. Thus, the main manifestation of hyperesthesia is pain that occurs as a reaction to sourness, to coldness, to hotness, to sweetness, to air flow, or during chewing. DH is considered a true pain syndrome [42].

The ideal desensitizing agent should not be irritable to the pulp; moreover, it should be relatively painless during application, easy to apply, and fast-acting, as well as likely have long-term or permanent effects [43,44]. As described by Godoi, F.A., et al. [43], different varnishes have a limited effect on the remineralization of the enamel surface, since, upon treatment, the surface micro-hardness values do not return to the physiological values. According to several authors, when using a high concentration of NaF (22,600 ppm F^−^), the varnishes allow a relevant remineralization of the most superficial layer of enamel, while minor effects occur on deeper layers [45,46,47].

In the present study, an electrometric technique was used to measure the desensitizing effectiveness of a calcium-fluoride-forming agent, Tiefenfluorid^®^, while Enamelast^TM^ was included as a control treatment. Our data provide direct evidence of the higher efficacy of Tiefenfluorid^®^ treatment in improving the overall condition of the DH patients with respect to the control group (treatment with Enamelast^TM^). In this respect, it is worth noting that the mechanism of Tiefenfluorid^®^ action (Figure 1) is complex; it involves the use of two different solutions, where the first one penetrates the enlarged interprismatic spaces of the enamel, while the second solution, consisting of Ca(OH)_2_, reacts with fluorides in the interprismatic spaces of the enamel. The result is the formation of very small crystals of calcium fluoride (CaF_2_), magnesium fluoride (MgF_2_), copper hydroxyfluoride (CuOHF), and silicate gel [36]. Furthermore, a deep fluoridation occurs because the CaF_2_ crystals are only 50 Å in size; because of this, they fit well into the dentinal tubules [36,48,49]. Moreover, in combination with very small MgF_2_ crystals and fluoride crystals, an optimal concentration of fluoride ion can be achieved. Hence, it can be assumed that the high concentration of fluoride ions, together with calcium, phosphate, and hydroxyl ions present in the saliva, can allow a strong physiological remineralization. Likely, very small crystals (CaF_2_ and MgF_2_), embedded in the silica gel, remain for a long time in the depths of the dentinal tubules [36].

It has been established that the reaction of fluoride present in varnishes with enamel depends on time. In particular, the concentration of soluble fluoride is relevant in the short term, while the dissolution of the insoluble fluoride, present in the varnish matrix, is critical at later timepoints [46]. Thus, in a short period of time, such as after a professional application, the concentration of soluble fluoride is responsible for the chemical formation of calcium-like fluoride reservoirs on the enamel; in contrast, the insoluble fluoride, included in the varnish formulation, plays a significant role in slowing down the abovementioned process [46]. Therefore, according to Godoi, F.A., et al. [43], the concentrations of soluble and insoluble fluoride should allow the formation of calcium-like fluoride deposits; in that study, Enamelast^TM^ was used at a higher fluoride concentration than other varnishes. In any case, after the application of the varnish fluoride, we should assume that the levels of fluoride in saliva vary, as also stated by different manufacturers, because of the variations of several parameters such as exposure time, fluoride retention, and/or fluoride clearance from the oral cavity [43].

The control group, treated in our study with Enamelast^TM^, showed that the level of sensibility was normalized rapidly after the treatment and persisted up to 3 months later, while a condition of hypersensitivity was again observed at 6 months. The increase in electro-sensitivity, measured by the pulp tester, was in line with the clinical data, since such patients reported the resurgence of DH symptoms (data not shown). Indeed, the patients subjectively affirmed the return of sensitivity, and the instrument objectively measured the augmented electro-sensitivity. Analysis of the dentin electro-sensitivity allowed us to conclude that, in all DH cases, the treatment with Tiefenfluorid^®^ returned the electro-sensitivity parameters to normal levels immediately after remineralization treatment, thus significantly affecting hypersensitivity.

When comparing the two treatment modalities, it should also be underlined that Tiefenfluorid^®^ requires a lower number of visits and applications than the Enamelast^TM^ varnish (three vs. seven), thus resulting in a simplified procedure and likely higher patient compliance.

An apparent limitation of the present study is the absence of a negative control group (untreated patients). As mentioned above, this is because we considered it unethical to withhold any treatment from patients with DH for such a long observation time. 

## 5. Conclusions

The treatment with Tiefenfluorid^®^ led to a greater reduction in DH than Enamelast^TM^ varnish, in adult patients with non-carious lesions. Furthermore, Tiefenfluorid^®^ provided long-lasting results that consistently improved at least up to 6 months (last follow-up timepoint), while the effect of the Enamelast^TM^ varnish was less evident and started to decrease after 3 months.

## Figures and Tables

**Figure 1 materials-15-01266-f001:**
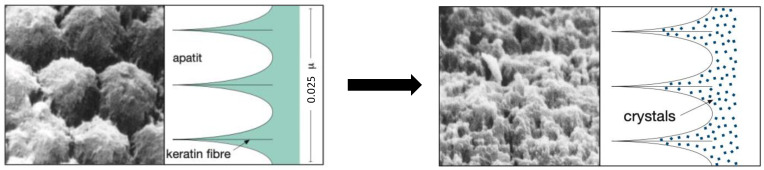
Mechanism of action of the two Tiefenfluorid^®^ solutions on loose enamel [36]. As depicted in the left figure, the application of the first solution allows the complex fluoride and copper ions to enter deeply into the interprismatic enamel. The right figure shows the homogeneous filling occurring in the interprismatic enamel by the second solution application.

**Figure 2 materials-15-01266-f002:**
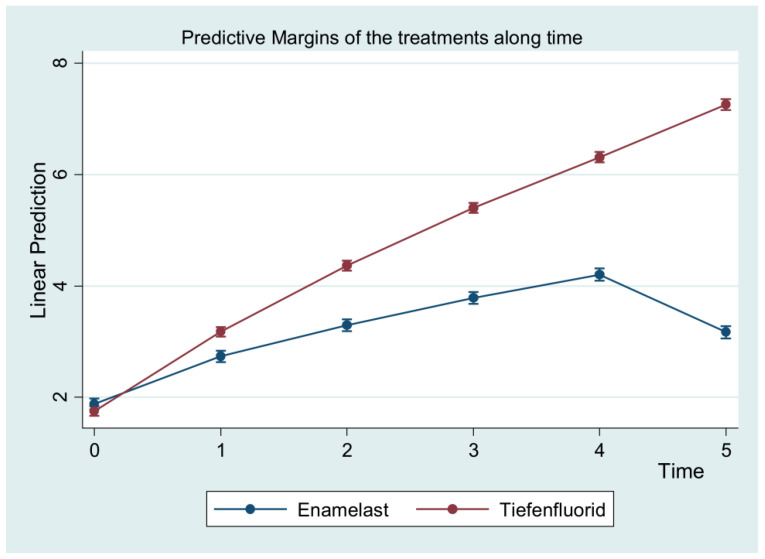
Predictive margins with 95% confidence intervals of DH pulp tester measurements at different timepoints.

**Table 1 materials-15-01266-t001:** Characteristics of the two fluoride-based agents.

Material Description	Manufacturer	Trade Name	Chemical Composition (Ingredients)
Calcium-fluoride-forming solution	Humanchemie GmbH, Alfeld, Germany	Tiefenfluorid^®^	1st solution: magnesia fluoro silicate (F_18_Mg_16_Na_10_O_66_Si_27_), copper (II) fluoro-silicate (CuF_6_Si), sodium fluoride (NaF) as stabilizer, distilled water 2nd solution: calcium hydroxide (Ca(OH)_2_)—highly dispersed, methylcellulose (C_6_H_7_O_2_(OH)x(OCH_3_)y), distilled water
Fluoride varnish	Ultradent Inc., Cologne, Germany	Enamelast^TM^	flavored, xylitol-sweetened, 5% NaF (22,600 ppm F^−^) resin

**Table 2 materials-15-01266-t002:** Pulp tester measurements in the two groups.

Time-Point	Treatment Group		Control Group	
	Mean	SD	Mean	SD
Before treatment	1.75	0.72	1.88	0.63
7 days	3.17	0.62	2.74	0.63
14 days	4.36	0.54	3.30	0.67
1 month	5.4	0.58	3.79	0.73
3 months	6.31	0.73	4.21	0.81
6 months	7.25	0.81	3.17	0.50

**Table 3 materials-15-01266-t003:** Random intercept/random slope model of the effect of treatment, time, and defect diagnosis on the measured dentinal sensibility.

Variable	Coefficient	Standard Error	z	p	95% Confidence Interval	
					Lower Bound	Upper Bound
Treatment	0.81	0.17	4.77 **	<0.001	0.48	1.15
Enamelast* hard tissue erosion	0.18	0.13	1.32	0.189	−0.09	0.44
Enamelast* cuneiform defect	0.17	0.14	1.22	0.222	−0.11	0.45
Enamelast* pathological abrasion	0.11	0.15	0.70	0.481	−0.19	0.40
Tiefenfluorid* gingival recession	−0.15	0.13	−1.21	0.225	−0.40	0.09
Tiefenfluorid* hard tissue erosion	−0.05	0.12	−0.40	0.690	−0.29	0.19
Tiefenfluorid* cuneiform defect	0.06	0.13	0.44	0.662	−0.20	0.31
Enamelast* 7 days	−0.09	0.03	−2.72 **	0.007	−0.15	−0.02
Enamelast* 14 days	−0.47	0.05	−9.53 **	<0.001	−0.57	−0.38
Enamelast* 1 month	−0.93	0.07	−13.33 **	<0.001	−1.06	−0.79
Enamelast* 3 months	−1.45	0.09	−16.11 **	<0.001	−1.63	−1.28
Enamelast* 6 months	−3.43	0.11	−30.84 **	<0.001	−3.65	−3.22
Tiefenfluorid* before treatment	−0.78	0.10	−8.12 **	<0.001	−1.00	−0.60
Tiefenfluorid* 7 days	−0.30	0.08	−4.00 **	<0.001	−0.45	−0.15
Tiefenfluorid* 14 days	−0.06	0.06	−1.08	0.281	−0.17	0.049
Tiefenfluorid* 1 month	0.03	0.04	0.90	0.367	−0.04	0.10

** Statistically significant at *p* < 0.01. Wald χ^2^ = 57,878.45, *p* ≤ 0.001. Enamelast* gingival recession, Tiefenfluorid* cuneiform defect, Enamelast* before treatment, Tiefenfluorid* 3 months, and Tiefenfluorid* 6 months were omitted because of collinearity.

**Table 4 materials-15-01266-t004:** Wald test for predictive margins of DH measurements stratified by time.

Variables	Contrast	SE	z	p	95% Confidence Intervals	
					Lower Bound	Upper Bound
Tiefenfluorid vs. Enamelast before treatment	−0.13	0.07	−1.92	0.055	−0.26	0.00
after 7 days	0.44 **	0.07	6.45	<0.001	0.31	0.57
after 14 days	1.07 **	0.07	15.39	<0.001	0.93	1.21
after 1 month	1.62 **	0.07	22.71	<0.001	1.48	1.76
after 3 months	2.11 **	0.07	28.86	<0.001	1.97	2.25
after 6 months	4.09 **	0.07	54.28	<0.001	3.94	4.24

** Statistically significant at *p* < 0.01.

## Data Availability

The data presented in this study are available on request from the corresponding author.

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
