# Peer review of "Effectiveness of Two Different Fluoride-Based Agents in the Treatment of Dentin Hypersensitivity: A Prospective Clinical Trial"

_materials, 2022, doi:10.3390/ma15031266_

Round 1

Reviewer 1 Report

No Comments

Author Response

Thank you to the Referee for the positive consideration about our revised paper!

Reviewer 2 Report

With the changes that the authors did the article has improved and as such I consider it to be published. 

Author Response

Thank you to the Referee for the positive consideration and to recommend our paper for publication!

Reviewer 3 Report

The current version of this paper is much improved, compared to the previous one.

My comments are as follows:

Lines 149-154. I understand that this information explains lines 145-147, but, in my opinion, the place in somewhere else, maybe in the introduction.

Lines 166-167. Please rephrase or remove the possible mechanism. Either is certain, either is not.

Scheme 1 should be Figure 1.

Figure 1 should be replaced with a much clear one.

Author Response

Reviewer 3_Round 1

Point 1: Lines 14-154. I understand that this information explains lines 145-147, but, in my opinion, the place in somewhere else, maybe in the introduction.

Answer 1: Thank you to the referee. The information is now settled in Introduction section, lines 63-70, with the respective references.

Point 2: Lines 166-167. Please rephrase or remove the possible mechanism. Either is certain, either is not.

Answer 2: Thank you for the advice. We have just removed “possible” word from the text and title in lines 175 & 178.

Point 3: Scheme 1 should be Figure 1.

Answer 3: Thank you for the comment. We changed the name in lines 176, 178 & 253.

Point 4: Figure 1 should be replaced with a much clear one.

Answer 4: Thank you to the referee. We replaced the Figure 2 (in the revised version) with a clear one (line 226).

Reviewer 4 Report

The authors have done the necessary correction with significant improvement of the manuscript. It can be accepted in the current form. But minor English correction is required.

Author Response

Reviewer 4_Round 1

Point 1: The authors have done the necessary correction with significant improvement of the manuscript. It can be accepted in the current form. But minor English correction is required.

Answer 1: Thank you to the referee for the positive consideration after the last review! As requested, the English was again carefully revised throughout the document.
